# A Review of Defatting Strategies for Non-Alcoholic Fatty Liver Disease

**DOI:** 10.3390/ijms231911805

**Published:** 2022-10-05

**Authors:** Erin Nicole Young, Murat Dogan, Christine Watkins, Amandeep Bajwa, James D. Eason, Canan Kuscu, Cem Kuscu

**Affiliations:** Transplant Research Institute, James D. Eason Transplant Institute, Department of Surgery, College of Medicine, The University of Tennessee Health Science Center, Memphis, TN 38163, USA

**Keywords:** liver, steatosis, defatting, molecular biology

## Abstract

Non-alcoholic fatty liver disease is a huge cause of chronic liver failure around the world. This condition has become more prevalent as rates of metabolic syndrome, type 2 diabetes, and obesity have also escalated. The unfortunate outcome for many people is liver cirrhosis that warrants transplantation or being unable to receive a transplant since many livers are discarded due to high levels of steatosis. Over the past several years, however, a great deal of work has gone into understanding the pathophysiology of this disease as well as possible treatment options. This review summarizes various defatting strategies including in vitro use of pharmacologic agents, machine perfusion of extracted livers, and genomic approaches targeting specific proteins. The goal of the field is to reduce the number of necessary transplants and expand the pool of organs available for use.

## 1. Introduction

The liver is a complicated organ, and one of its main functions is to regulate lipid homeostasis through the interplay of various hormones, nuclear receptors, and transcription factors [1]. This includes conducting synthesis of new lipids, coordinating their transport to other parts of the body, and utilizing them as energy substrates [2]. Any imbalance between these pathways can result in the accumulation of fat. In the absence of alcohol consumption, non-alcoholic fatty liver disease (NAFLD) is characterized by the build-up of ectopic fat in the liver, also known as steatosis [3]. NAFLD is the current liver pandemic sweeping the globe. It is estimated that about 25% of the world’s population is affected by NAFLD [4]. There is a very close association with dyslipidemia, type 2 diabetes, central obesity, and metabolic syndrome, each with an occurrence of 69%, 23%, 51%, and 43%, respectively [4]. Consequently, the rate of NAFLD diagnoses has been steadily rising with the ever-increasing obesity burden. Although the prevalence of this disease is so high, public knowledge of its existence and effects is still lacking.

More information is constantly being revealed on the pathophysiology of NAFLD, however, the general concept is that more lipids are being retained than the hepatocyte is able to expel. The four major pathways regulating lipid acquisition and disposal include uptake of free circulating fatty acids, de novo lipogenesis, fatty acid oxidation and transportation of lipid to the outside of liver as very low-density lipoproteins (VLDL) [5] (Figure 1). Focusing on the uptake of circulating lipids, it is directly proportional to the concentration of plasma free fatty acids (FFAs) that are mainly derived from the body’s supply of adipose tissue [6]. This mechanism will be explored more in terms of genetic modifications that can be made to lipid transport proteins. Lipid accumulation can be characterized as either microsteatosis or macrosteatosis. Microsteatosis represents small lipid droplets that do not displace the cell’s nucleus, and these livers generally are not associated with many complications. Macrosteatosis, which involves displacement of the nucleus, is a condition that can range from simple lipid build-up with minimal effects to non-alcoholic steatohepatitis (NASH) that could result in fibrosis, cirrhosis, and eventual end-stage liver disease [3]. With these complications in mind, one can imagine how NAFLD has become a leading cause for liver transplantation as well as the reason for many livers not being suitable for donation. Donor liver steatosis is a significant risk factor for post-transplant complications due to its increased susceptibility to ischemia reperfusion injury (IRI) which could result in post-operative graft dysfunction, graft loss and requirement for re-transplantation [7].

Regardless of how fat accumulates in the hepatocytes, its presence can result in endoplasmic reticulum, oxidative, and mitochondrial stress along with impaired autophagy [3]. The two-hit hypothesis was proposed in 1998 by Day and James to explain the pathogenesis of NAFLD [8]. According to their model, the “first hit” occurs when triglycerides (TGs) build-up in the hepatocytes. Inflammation and necrosis follow in the lipid-filled hepatocytes when the TGs undergo peroxidation, thus signifying the “second hit”. With the progression of more cellular damage, NAFLD can evolve into NASH that is characterized by increased amounts of inflammation. Of the patients with NASH, approximately 5–18% develop cirrhosis, and when combined with fibrosis, about 38% develop cirrhosis [9]. Patients with NASH and fibrosis or cirrhosis are at an increased risk of hepatocellular carcinoma.

A large amount of research has gone into understanding the pathophysiology of NAFLD in order to find ways to prevent and possibly reverse fat accumulation in hepatocytes. This paper will review current in vitro defatting techniques and machine perfusion therapy along with exploring possible genomic approaches that could affect fatty acid transport into hepatocytes as well as de novo lipogenesis. Targeting specific genes and proteins involved in the mechanism of steatosis could be a potential avenue of future treatment for both donors and those on the transplant list.

## 2. In Vitro Defatting Techniques

Over the years, there have been several studies conducted to determine the effects of in vitro defatting strategies. In 2013, Nativ et al. explored the effect of macrosteatosis reduction approaches on lipid droplet size as well as hepatocyte viability and functions specific to the liver [10]. In this experiment, primary hepatocytes from lean Zucker rats were cultured in a medium with equal amounts of both oleic and linoleic acids to induce steatosis [10,11,12]. Then, forskolin (a glucagon mimetic), the PPARα agonists (GW7647, GW501516), scoparone, hypericin (a pregnane X receptor [PXR] ligand), visfatin (an adipokine), and amino acids were used as a cocktail for defatting purposes [9]. This cocktail had been shown to reduce macrosteatosis by activating hepatocellular TG metabolism [12]. This study showed that macrosteatosis can be inducible in primary hepatocytes. They also confirmed that macrosteatosis can be reversable with SRS (steatosis reduction supplements). They also concluded that accelerated macrosteatotic reduction led to a faster recovery of urea secretion and bile canalicular formation with the same viability as seen in lean rat hepatocytes. These results indicate that hepatocyte functional recovery is dependent on either macrosteatosis reduction time or direct effects of the SRS.

Nativ et al. conducted another study to determine the sensitivity of hepatocytes to hypoxia/reoxygenation (H/R) stress after exposure to defatting protocol [13]. They used the same protocol including SRS enriched with L-carnitine. They showed that the amount of steatosis is correlated with vulnerability to H/R injury, and lean or microsteatotic hepatocytes are resistant to injury. These macrosteatotic hepatocytes have less ability to produce ATP and that is related to increased production of reactive oxygen species. Lean and microsteatotic hepatocytes also showed improved activity in liver function like urea secretion and bile canalicular transport compared to those with macrosteatosis. They recommended that the lipid lowering defatting agents and combinations could be beneficial to overcome H/R stress and provide a possible recovery mechanism for discarded macrosteatotic liver grafts [13].

Another experiment conducted in 2016 by Yarmush et al. examined the use of a defatting cocktail, as described by Nativ et al. [10,13], on HepG2 cells with induced steatosis [14]. They performed their experiment under normoxic and hyperoxic conditions. They showed that TG storage levels decreased along with an increase in beta-oxidation, the tricarboxylic acid cycle, and the urea cycle. All these parameters were augmented by SRS and hyperoxic conditions. They also found that the rate of extracellular glucose uptake was miniscule compared to the amount supplied by glycogenolysis within the cell which has also active glycolysis. In conclusion, both glycolysis and beta-oxidation were occurring at the same time, which is not expected under normal conditions. Typically, glycolysis occurs when the body is in the fed state while beta-oxidation occurs during fasting. However, the combination of defatting agents does not mimic any known metabolic conditions, so the unusual result could be anticipated. 

In 2018, Boteon et al. focused to determine the effect of defatting agents on primary human liver cells [15]. This was the first study to use primary human hepatocytes (PHH), which were isolated from discarded donor livers in this type of experiment. PHH cells were incubated with standard medium supplemented with FFAs, consisting of palmitic, linoleic, and oleic acids. Forskolin, GW7647, hypericin, scoparone, GW501516, visfatin, and L-carnitine were added and incubated for 48 h. They successfully reduced the intracellular lipid accumulation by 54% and TG levels by 35%. Furthermore, production of ketone bodies was increased, indicating that beta-oxidation was occurring at a higher rate. In cytotoxicity experiments, they used human intra-hepatic endothelial cells (HIEC) and human cholangiocytes, and the viability was measured for all cultures, including PHH. They demonstrated an 11% increase in viability of PHH treated with defatting drugs compared to the fatty control group. Moreover, there was no difference in viability between the treated and control groups of HIEC while the viability of human cholangiocytes improved, although this result was not statistically significant. It was the first study to prove that defatting drug cocktails have efficacy in reducing lipid content of PHH while causing no harm to non-parenchymal cells.

Recently, Aoudjehane and colleagues used a novel defatting cocktail in 3 different human culture models: PHH with induced steatosis, PHH isolated from a steatotic liver, and precision-cut liver slices (PCLS) from a steatotic liver [16]. They used a similar defatting cocktail (including forskolin, L-carnitine, and a PPARα agonist GW7647) in addition to two new agents, rapamycin and necrosulfonamide. Rapamycin is an immunosuppressant that can reduce steatosis by inhibiting mammalian target of rapamycin (mTOR). This action promotes lipogenesis, TG secretion, and macro-autophagy [17,18,19]. Necrosulfonamide (NSA) is an inhibitor of an effector in the necroptosis pathway that recently was revealed as a regulator of TG storage in the liver [20]. The new cocktail showed a significant decrease in lipid droplets and TG levels in steatosis-induced PHH, PHH isolated from fatty livers and also in PCLSs. Additionally, they reported a reduction in endoplasmic reticulum stress and reactive oxygen species production. By using PCLS, this study was the first to demonstrate defatting agents are successful in a model that maintains the 3D structure of liver tissue and the interactions between hepatocytes and other liver cell types. Table 1 summarizes chemicals that have been used in in vitro experiments.

## 3. Machine Perfusion Defatting Techniques 

### 3.1. Preclinical Studies 

The detrimental effects of static cold storage on transplantable livers have produced a need for a more efficient strategy to store organs for future use, such as ex vivo machine perfusion. Ex vivo machine perfusion has been shown to not only reduce storage injury, but also provides an opportunity to treat damaged livers before transplantation. Much research has gone into this field over recent years leading it to become an alternate strategy to reduce preservation injury [21]. 

Bessems et al. conducted a study to compare cold storage versus machine perfusion [21]. They induced macrosteatosis in a rat model by feeding it a methionine and choline-deficient diet before using either hypothermic cold storage or machine perfusion (4 °C) for 24 h. They reported that machine perfused livers had significantly less damage as well as higher bile production, ammonia clearance, urea production, oxygen consumption, and ATP levels compared to static cold storage samples. Additionally, Kron et al. tested similar experiment to determine the viability of machine perfused grafts after transplantation [22]. After inducing macrosteatosis, livers were transplanted after either <1 h of cold storage, 12 h of cold storage, 12 h of cold storage followed by 1 h of hypothermic oxygenated perfusion (HOPE) or 12 h of cold storage followed by 1 h of hypothermic nonoxygenated perfusion (HNPE). They reported that HOPE therapy before transplantation resulted in significantly decreased reperfusion injury evidenced by less oxidative stress, nuclear injury, macrophage activation and fibrosis after one week. However, these protective effects were lost with the absence of oxygen in the perfusate, and this study did not find any reduction in the level of steatosis after HOPE therapy. Early trials of normothermic (37 °C) machine perfusion with an oxygenated blood-based perfusion system showed that a reduction in steatosis could be achieved in porcine models [23]. Even without the addition of a pharmacologic defatting cocktail, Jamieson et al. observed a 13% reduction in diet-induced liver steatosis after 48 h of normothermic ex vivo machine perfusion. However, other studies have since shown that the use of defatting cocktails further enhances the effect of machine perfusion.

Nagrath et al. showed results describing a 65% reduction in hepatocyte TG content after normothermic perfusion with a defatting cocktail (PPARα ligand (GW7647), a PPARδ ligand (GW501516), hypericin, scoparone, forskolin and visfatin) for three hours [12]. They concluded that more oxygen availability led to an increase in beta-oxidation and subsequent reduction in steatosis. Although, normothermic perfusion alone still reduced TG content by 30%, demonstrating the inherent defatting capabilities of this technique. They further tested if subnormothermic (20 °C) perfusion with the same defatting cocktail [12] would have an equivalent effect while being easier to maintain [24]. The results revealed no significant reduction in intracellular lipid content after six hours of perfusion in livers from obese Zucker rats. This indicates that higher temperatures may be required for cellular metabolism, specifically for beta oxidation, to take place at a high rate.

Another novel mechanism for liver defatting was described by Vakili et al. and involves the use of glial cell line-derived neurotrophic factor (GDNF) [25]. They had previously explained the mechanism behind the protective effect of GDNF against high-fat diet-induced steatosis in mice by reducing PPARγ expression [26] and wanted to test the potential of it before transplantation. Steatotic and lean donor livers were both perfused with either the vehicle, GDNF, or the same defatting cocktail described previously. They found that GDNF was equally effective as the defatting agents [12] at reducing TG content in hepatocytes (>40% reduction); however, GDNF induced less liver damage than the defatting cocktail, indicated by a significant rise in lactate dehydrogenase activity. Moreover, GDNF may prove to be a more suitable option for treating livers ready for transplant.

In a recent study, Raigani et al. demonstrated how a defatting cocktail in normothermic perfusion improves markers of cell viability in mouse model [27]. They implemented the same defatting cocktail supplemented with L-carnitine and amino acids [12]. They showed a reduction in perfusate lactate and better bile quality along with a decrease in inflammatory markers such as tumor necrosis factor-α (TNFα), NF-κB, and apoptosis markers, specifically caspase-3 and Fas cell surface death receptor. The results also showed an increase in gene expression of mitochondrial beta-oxidation markers; however, there was not a significant reduction in hepatocyte steatosis. They concluded that perhaps a clinical improvement in liver function is more important than decreasing lipid content alone. Table 2 summarize the preclinical studies.

### 3.2. Clinical Trials

Although much work has been done dealing with fatty livers in animal models, researchers are still working out the kinks when conducting studies on human livers. In 2010, Guarrera and colleagues first experimented with human livers to see if hypothermic machine perfusion (HMP) would preserve the organs better than cold storage, and they found that HMP is promising to use before transplantation and may improve graft function depending on early biochemical markers [28]. Other studies have looked at using hypothermic machine perfusion to assess the quality of liver grafts [29,30]. Both groups used human livers, some potentially transplantable and some discarded mainly due to steatosis. The studies found that damaged livers released higher levels of injury markers such as AST, ALT, and lactate dehydrogenase (LDH). Discarded donor livers also had a lower ATP recovery rate compared to the potentially transplantable group. At hypothermic temperatures, both groups also found that the morphology was preserved with or without oxygen supplementation.

Clinical trials using HOPE to preserve livers destined for transplant have also been underway in recent years. These studies evaluated liver function in transplant patients after their organ was preserved using HOPE or static cold storage. One study specifically looked at the risk of non-anastomotic biliary strictures (NAS) and found that livers preserved with HOPE had significantly less occurrences of biliary strictures, post reperfusion syndrome, and early allograft dysfunction [31]. Two other studies have also used HOPE to preserve grafts but observed differences in liver enzymes and function after transplantation [32,33]. Both groups found that patients who received livers preserved using HOPE had significantly lower levels of liver injury enzymes, graft dysfunction, 90-day complications and hospital stay. These results are very promising for the future of liver transplantation.

A great deal of research has also gone into the effects of normothermic machine perfusion on liver graft quality. There have been several studies specifically using normothermic perfusion to assess graft viability by evaluating markers like injury enzymes, lactate clearance, and bile production [34,35,36]. These studies were performed on previously discarded livers and determined that they were suitable for transplant after normothermic perfusion. Grafts perfused before transplant typically resulted in better patient outcomes and less post-surgery complications. There has also been more interest in normothermic perfusion for treating hepatic steatosis. For example, Liu et al. investigated changes in the lipid profiles on ten perfused livers [37]. They found that perfusate TG levels significantly increased from 1-h to 24-h of treatment; however, there was a decrease in perfusate levels of total cholesterol, high-density lipoproteins, and low-density lipoproteins. Additionally, there was no significant decrease in steatosis histologically. They believed their findings to be due to differences in hepatic morphology compared to animal models along with the chronic accumulation of fat in humans compared to diet-induced steatosis in animal experiments. Although the increase in perfusate TGs indicates that active metabolism occurred in the grafts. Moreover, normothermic machine perfusion may be a potential starting point for liver defatting if combined with pharmacological cocktails.

Boteon and colleagues conducted an experiment using normothermic machine perfusion supplemented with a previously described defatting cocktail [12] to assess intracellular lipid reduction in ten discarded livers [38]. Compared to a control group perfused with vehicle only, the five livers perfused with defatting agents for six hours had a reduction in tissue TGs and macrosteatosis by 38% and 40%, respectively. The team also saw increased beta-oxidation with higher ATP production along with enhanced viability markers such as urea production, bile production, and lowered injury enzymes. With such promising results, one could get excited about the future of pre-transplant liver treatment. Even though the same group has previously demonstrated minimal cytotoxicity of these drugs directly on cholangiocytes and intrahepatic endothelial cells [16], the systemic effects have not yet been investigated in humans. More extensive trials would need to take place to establish drug safety before this defatting strategy could be implemented in patients. In Table 3, perfusion methods in clinical trials have been summarized.

## 4. Genomic Approaches for Defatting Strategies

Hepatic steatosis is the result of dysfunction between the four major pathways regulating lipid metabolism in the liver. These pathways include the uptake of fatty acids (FAs), de novo lipogenesis, beta-oxidation, and transport out of hepatocytes [5]. Abnormal levels of proteins associated with both the uptake and transport of lipids and de novo lipogenesis have been documented in patients with NAFLD, and these provide a potential target for treatment. Specifically, this section will focus on recent studies that have incorporated genomic approaches to influence the processes involved in intracellular lipid accumulation.

To begin with hepatocyte lipid transport, FAs are primarily moved into the cells by transporters with diffusion playing a much smaller role [39]. The main proteins involved in this process include fatty acid transport proteins (FATP), cluster of differentiation 36 (CD36), and caveolins, all located in the plasma membrane. Out of six possible FATPs, FATP2 and FATP5 are primarily found in the liver and have been implicated in the pathogenesis of NAFLD [40]. Studies have shown that both FATP2 and FATP5 are expressed at higher levels in patients with NAFLD that progresses to NASH [5]. 

Falcon et al. conducted a study to characterize the role of FATP2 in lipid transport [41]. By using an adeno-associated virus (AAV)-based knockdown strategy, they were able to inhibit the function of FATP2 in vivo using mouse models. One week after injection with the viruses, a 40% decrease in intracellular lipid uptake was observed in the FATP2 knockdown mice compared to control. The team also found that knockdown of FATP2 resulted in lowered liver TGs but did not affect hepatic free and esterified cholesterol levels. Liver injury enzymes, liver histology, and feeding behavior were also not affected by inhibition of FATP2, indicating this as a promising route for further investigation. Additionally, early studies also using AAV-based strategies showed that knockout of FATP5 also significantly reduces FA uptake in hepatocytes, TG content, and reverses steatosis [42,43]. 

CD36 is also found at higher levels in patients with NAFLD. CD36 is a translocase protein that aids in the transport of long-chain FAs and is regulated by PPAR-γ, pregnane X receptor, and liver X receptor [44]. Wilson et al. determined the role of CD36 in the pathogenesis of NAFLD [45]. They found that deletion of the *Cd36* gene in mice fed with a high-fat diet resulted in reduced liver lipid content and hepatocyte FA uptake. Additionally, the mice had improved whole-body insulin sensitivity and reduced liver inflammatory markers, making this protein an excellent target for NAFLD gene therapy.

The caveolin protein family consists of three members, termed caveolins-1, 2, and 3, found in the plasma membrane and whose function is to facilitate protein trafficking and lipid droplet formation [40]. Early studies found that caveolin-1 levels were increased in mice fed a steatosis-inducing high-fat diet for 14 weeks, indicating that caveolins may be implicated in the pathogenesis of NAFLD [46]. In contrast, recently, Li et al. demonstrated that wild-type mice fed a high-fat diet with NAFLD had markedly reduced expression of the caveolin-1 gene [47]. Similarly, mice with caveolin-1 knockdown had augmented steatosis, increased plasma cholesterol, and elevated liver injury enzymes, whereas overexpression of the gene resulted in significantly attenuated lipid accumulation in hepatocytes. Another study aimed to determine the protective mechanism behind caveolins in NAFLD through both in vitro and in vivo methods [48]. Taken together, this study revealed decreased levels of caveolins and autophagy-related proteins when exposed to high levels of FAs for an extended period of time. They further concluded that the inhibition of Akt/mTOR pathway was involved in the protective role of caveolin-1 in autophagy and lipid metabolism in NAFLD. 

Fatty acid binding protein (FABP) is another lipid transport protein that is found intracellularly. Following passage through the plasma membrane, lipids are not allowed to travel freely through the cytosol; instead, FABPs shuttle them between different organelles where various metabolic processes take place. Specifically, FABP1 is the predominant isoform found in the liver [40]. It is thought that FABP1 is cytoprotective due to its ability to facilitate the oxidation or TG incorporation of potentially lipotoxic FAs that accumulate in the cytosol [49]. An early study demonstrated that patients with NAFLD had much higher mRNA levels of FABP1 compared to controls [50]. The authors concluded that this was a compensatory mechanism where the cell attempted to store or release excess lipid. However, the enhanced FA trafficking may lead to storage of harmful lipid levels, promoting steatosis. Knocking out FABP1 in mice resulted in decreased hepatic TGs and lipid disposal pathways, as expected [51]. Additionally, a more recent study of FABP1 knockout found that expression of inflammatory and oxidative stress markers, as well as a marker of lipid peroxidation, were also significantly decreased [52]. Findings like these indicate that attenuation of FABP1 production could be a potential route of treatment for NAFLD.

De novo lipogenesis (DNL) enables the liver to synthesize new FAs from acetyl-CoA [5]. The two main enzymes involved are acetyl-CoA carboxylase (ACC) and fatty acid synthase (FASN). Once the new FAs have been synthesized, they must undergo a myriad of modifications before being stored as TGs or exported as VLDL particles [5]. Thus, an increased rate of DNL could easily cause accumulation of lipids inside hepatocytes, leading to steatosis and eventually NAFLD.

Two key transcription factors involved in the regulation of DNL include sterol regulatory element-binding protein1c (SREBP1c) and carbohydrate regulatory element-binding protein (ChREBP) [5]. Beginning with SREBP1c, it is activated by insulin and liver X receptor α [53]. Studies have found elevated levels of SREBP1c in patients with NAFLD [54] along with an expected rise in hepatic TG levels in mice genetically engineered to overexpress the protein [55]. Moreover, SREBP1c knockout mice displayed decreased levels of mRNAs encoding enzymes like ACC and FASN, both critical for DNL [56]. ChREBP, on the other hand, is only activated by carbohydrates and does not facilitate fat-induced lipogenesis, whereas high-fat diets may reduce the activity of ChREBP [5]. Lizuka et al. reported that ChREBP knockout in mice resulted in reduced hepatic TG synthesis by 65%, but insulin resistance, delayed glucose clearance, and intolerance to simple sugars was also observed [57]. This goes to show the importance of ChREBP in both DNL and glucose metabolism. Another experiment demonstrated that ChREBP knockout protected against fructose-induced steatosis in mice while enhancing hepatic damage through increased cholesterol synthesis and resultant cytotoxicity [58]. This finding leads the authors to believe that ChREBP may have a cytoprotective effect by limiting cholesterol toxicity. Thus, increased levels of ChREBP in patients with NAFLD may be due to a defense mechanism to prevent progression to NASH. It has also been postulated that lipogenesis may cause steatosis but prevents disease conversion to NASH [59]. Finally, hepatic overexpression of ChREBP in mice produced steatosis from upregulated DNL, but also maintained insulin sensitivity and glucose tolerance [60]. As detailed, both transcription factors of DNL play important roles in the pathogenesis of NAFLD, however, it was concluded that SREBP1c is the predominant regulator [60]. Patients with NAFLD had lowered levels of ChREBP while SREBP1c was upregulated, causing increased activity of ACC and FASN. 

As previously stated, ACC is a crucial enzyme that regulates the rate-limiting step of DNL and is elevated in response to SREBP1c in NAFLD patients. Early studies were conducted with knockout of different isomers of the enzyme. Mao et al. generated liver-specific ACC1 knockout mice where generation of malonyl-CoA, the product of ACC1, was 75% lower compared to control mice, and the livers accumulated 40–70% less TGs after feeding a fat-free diet for 10 days [61]. However, the synthesis of lipogenic enzymes was increased in ACC1 knockout livers, possibly due to overexpression of ACC2. Another study demonstrated that inhibition of both ACC1 and ACC2 achieved reversal of hepatic steatosis, reduced malonyl-CoA levels, and improved insulin sensitivity by increasing the rate of beta-oxidation [62]. Very recently, a great deal of work has been done to create small molecule inhibitors of ACC1 and 2. Matsumoto et al. used GS-0976 (fircosostat) in mice with diet-induced steatosis to inhibit both isoforms which resulted in significantly reduced TG hepatocyte content, histologically [63]. The small molecule also reduced the areas of hepatic fibrosis and treated high levels of liver injury markers, indicating a potential use in NASH as well as NAFLD. Another novel small molecule inhibitor specific for ACC1, called compound 1, has been investigated for its potential role in the treatment of NAFLD [64]. It has been shown that dual inhibition of ACC1 and 2 causes increase in plasma TG levels, which could be harmful to some patients [63]. However, selective ACC1 inhibition in mice did not affect plasma TG levels compared to controls while still achieving a significant reduction in hepatic steatosis and fibrosis. The study concluded that this effect is due to preserved activity of ACC2 that compensates for the loss of malonyl-CoA production from ACC1. Overall, these drugs show great promise in the future of NAFLD/NASH treatment.

FASN is another key enzyme involved in DNL whose primary role is converting malonyl-CoA to palmitate [5]. Similar to ACC, studies have also been done to show to effects of modifying the activity of FASN in hepatocytes. A study was conducted to evaluate the effects of FASN knockout in mice fed a zero-fat diet [65]. After prolonged fasting, the mice unexpectedly displayed hypoglycemia, fatty liver, and defects in PPARα expression which is a transcription factor integral to beta oxidation of FAs. Overexpression of FASN did not induce any histological changes in liver; however, complete genetic ablation of FASN resulted in a decline in cell proliferation and a rise in apoptosis [65]. Although inhibition of FASN may improve liver viability by decreasing DNL, it seems to have more important regulatory properties that are disastrous for the cell if interrupted. 

Another enzyme involved in liver lipid metabolism is stearoyl-CoA desaturase-1 (SCD1), and it is responsible for conversion of saturated Fas to monounsaturated Fas [5]. This effect is thought to be protective against NAFLD [60]. To corroborate this thought, Li et al. discovered that incubation of hepatocytes with saturated Fas lowered cell viability while incubation with monounsaturated Fas did not affect viability even though there was enhanced lipid accumulation [66]. Within the same study, SCD1 knockout was performed in mouse hepatocytes with diet-induced NASH and resulted in increased fibrosis and cellular apoptosis. Therefore, inhibition of SCD1 could exacerbate NAFLD or NASH due to excessive build-up of cytotoxic lipid species when monounsaturated Fas are not created to be safely stored. Overall, SCD1 proves to be a potential effective target for the treatment of NAFLD progression.

In a recent study, it has been shown that the transcription factor zinc fingers and homeoboxes 2 (ZHX2) also has a role in NASH models [67]. ZHX2 suppressed NASH progression in steatotic hepatic cells and downgraded inflammation and fibrosis in liver. Conversely, knocking out ZHX2 exacerbated NASH progression in animal models which was confirmed by increased lipid accumulation, aggravated inflammation, and increased fibrosis scores in liver. This protective effect is predominantly through activation of the phosphatase and tensin homolog (PTEN) gene by ZHX2.

In addition to the characterized genes and their functions in lipid accumulation in hepatocytes, functional genomics will help us find novel genes for defatting purposes. Recently, Hilgendor et al. used genome wide CRISPR screening for adipocytes and sorted the cells according to their lipid content. They identified candidate adipogenic regulators [68]. Similar experiments will shed light on hepatocyte lipid accumulation and will give us some novel targets to prevent fat accumulation in hepatocytes. 

Improvement of techniques and a more affordable cost in next generation sequencing have allowed researchers to accumulate more genetic information from patients. In this regard, genome-wide association studies (GWAS) will identify genes and their associated alterations in the genome (called SNPs) with a trait or disease like NAFLD. Two independent groups used large cohorts (10k and 20k) to find a novel association between NAFLD and target genes in European ancestry [69,70]. PNPLA3 has been characterized as a risk factor for the susceptibility of fat accumulation. Further studies in different geographic locations and with more diversity in race are necessary to conclude the gene-trait interaction. Table 4 summarizes the gene manipulations for liver defatting.

## 5. Additional Interventions for Liver Defatting

Both surgical and medical interventions have been shown to have an effect on people diagnosed with NAFLD. For example, bariatric surgery is an option for patients with obesity-related diseases who are able to commit to long-term medical follow-up. It has been shown that after bariatric surgery, patients have significant improvement in steatosis (resolution of NASH in 80% of patients in first year following surgery), which is related to the improvement in insulin resistance [71,72]. In addition to the resolution of NASH, decreasing fibrosis in the liver could be beneficial for the health of the liver [73]. However, some studies have controversial results regarding improvement in liver fibrosis in NASH patients [72]. In a recent long term follow-up study, NASH resolution in liver samples was observed in 84% of patients 5 years after bariatric surgery, and the reduction in liver fibrosis was most prominent in the first year with continued progression through 5 years [74].

Another option to possibly treat people with NAFLD includes the use of currently available medicines. One of the major drivers of NASH is insulin resistance as well as the dominant characteristics of type 2 diabetes and obesity. In addition to insulin resistance, dysfunction and dysregulation of adipose tissue leads to increased circulating fatty acids, carbohydrates and causes generation of hepatic lipid accumulation, cell injury on hepatocytes, increased inflammation, and liver fibrosis in the long run [75,76,77]. Glucagon-like peptide 1 (GLP-1) is an intestinal hormone that has a role in regulation of glucose metabolism. It stimulates insulin secretion, proinsulin gene expression and β-cell proliferative and anti-apoptotic pathways, as well as inhibiting glucagon release, gastric emptying, and food intake [78]. GLP-1 receptor agonists are a class of drugs that are beneficial to break the insulin resistance pattern of NAFLD. Liraglutide (one of the FDA approved GLP-1 receptor agonists) has been shown to improve liver function levels and reduce liver fat progression [79]. It has histologically been proved that it is beneficial for NASH resolution at liver tissue [80]. Another GLP-1 receptor agonist, semaglutide, is approved for the treatment of type 2 diabetes [81] and is being studied for use in weight management [82]. Semaglutide has a similar mechanism of action as liraglutide, but the metabolic effects of semaglutide are more robust [83,84,85]. Besides its effect on control of diabetes and weight loss, it reduces the AST levels and inflammation markers in liver [86]. In a recent randomized, placebo-controlled, phase 2 trial, it has been showed that semaglutide increased the NASH resolution compared to placebo in patients with biopsy-confirmed NASH and liver fibrosis. However, they did not show improvement in liver fibrosis [87].

## 6. Conclusions

NAFLD is an ever-growing problem affecting millions of people each day, but there is not currently a recommended treatment to prevent or cure this formidable disease. The increasing number of liver transplants due to NAFLD has led to a huge need for suitable donor livers to be available. In addition, many potential grafts are discarded because they are also too steatotic for successful transplantation. Much research has focused into defatting strategies to treat those living individuals with NAFLD and techniques to generate suitable donor grafts for transplantation. Medical and surgical interventions, targeting specific proteins or identification of novel target genes for defatting purposes, defatting using pharmacologic agents, and machine perfusion of extracted livers are all exciting avenues to explore for future treatment of this disease (Figure 2). However, issues are likely to arise with any method due to the extremely interconnected physiology of the liver where one change could have many effects far from the intended result. Therefore, all possibilities must be taken into careful consideration when developing treatments for NAFLD.

## Figures and Tables

**Figure 1 ijms-23-11805-f001:**
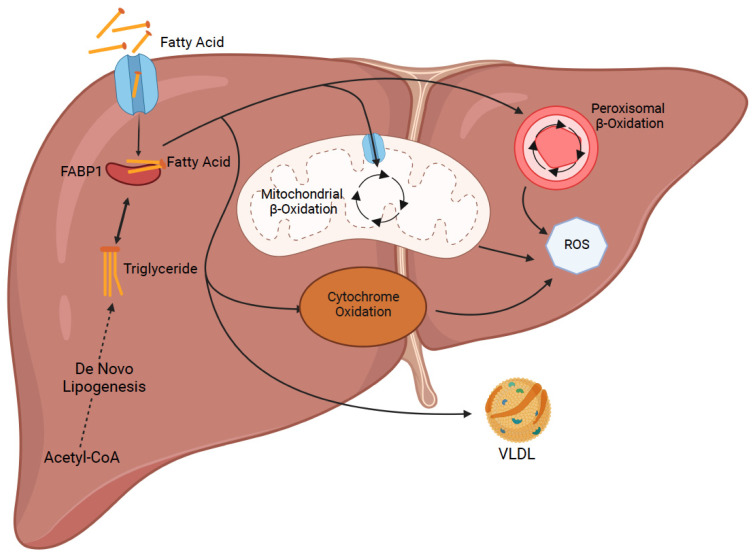
Major pathways regulating lipid acquisition and disposal inside the liver. Uptake of free circulating lipids and de novo lipogenesis increase the amount of fat in liver cells. Several fatty acid oxidation mechanism and transport of fat as low density lipoproteins (VLDL) decrease the amount of fat inside the liver cells (Created with BioRender.com).

**Figure 2 ijms-23-11805-f002:**
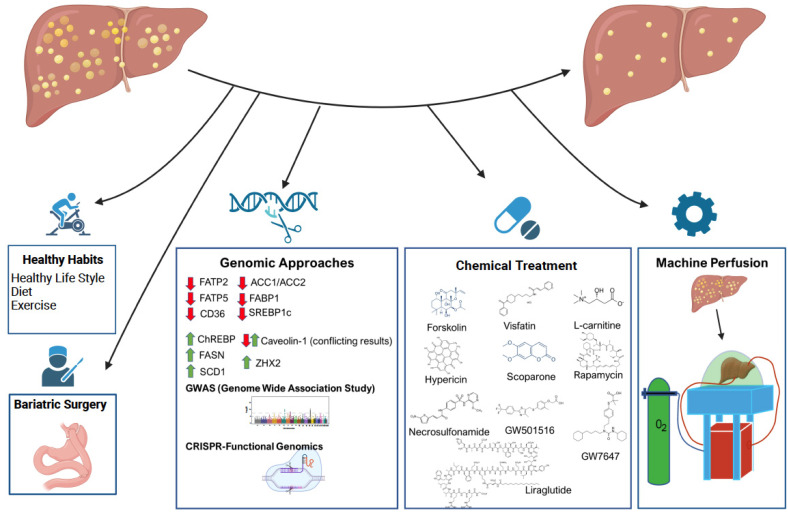
Overview of the different approaches for defatting purposes. Red arrow shows the beneficial effect of down regulation of target genes while green arrow demonstrates the positive effect of upregulation on defatting in third panel. Fourth panel summarizes the structure of known chemicals from in vitro defatting experiment. Perfusion of liver on normo- and hypothermic machine has great premise for future therapeutic approaches (Created with BioRender.com).

**Table 1 ijms-23-11805-t001:** Summary of In Vitro Defatting Techniques.

First Author	Year	In Vitro Model	Defatting Strategy	Effects of Agents
Mao et al. [10]	2013	Rat hepatocytes	Forskolin, PPARα and δ agonist, scoparone, hypericin, visfatin, amino acids	Faster steatosis reduction; recovery of urea secretion and bile canalicular formation
Nativ et al. [13]	2014	Rat hepatocytes	Forskolin, PPARα and δ agonist, scoparone, hypericin, visfatin, amino acids ± L-carnitine ± 90% O_2_	Higher reduction in TGs, increase in β-oxidation and ATP levels with L-carnitine and hyperoxia
Yarmush et al. [14]	2016	Human hepatoma cells	Forskolin, PPARα and δ agonist, scoparone, hypericin, visfatin, amino acids ± 90% O_2_	Decreased TGs, increased β-oxidation, TCA cycle and urea cycle, especially with hyperoxia
Boteon et al. [15]	2018	PHH, HIEC, human cholangiocytes	Forskolin, PPARα and δ agonist, scoparone, hypericin, visfatin, L-carnitine	PHH-decrease in lipids and TGs; increased viability of PHH and cholangiocytes; no cytotoxic effects on HIEC
Aoudjehane et al. [16]	2020	PHH, PHH from human fatty liver, human PCLS	Forskolin, L-carnitine, PPAR α and δ agonist, rapamycin, necrosulfonamide	Decrease in lipids and TGs and endoplasmic reticulum stress and production of reactive oxygen species

**Table 2 ijms-23-11805-t002:** Summary of Machine Perfusion Techniques–Preclinical Studies.

First Author	Year	Temperature of Perfusion	Additional Agents	Effects of Perfusion
Bessems et al. [21]	2007	Hypothermic	None	Less cell damage; increased bile production, ammonia clearance, urea production, O_2_ consumption, and ATP levels
Kron et al. [22]	2017	HOPE	None	HOPE: less oxidative stress, nuclear injury, macrophage activation and fibrosis; no decrease in steatosis
		HNPE		HNPE: loss of protective effects seen with HOPE therapy
Jamieson et al. [23]	2011	Normothermic	None	13% reduction in steatosis
Nagrath et al. [12]	2009	Normothermic	PPARα and δ ligands, hypericin, scoparone, forskolin and visfatin	65% reduction in TG content
Liu et al. [24]	2013	Subnormothermic	PPARα and δ ligands, hypericin, scoparone, forskolin and visfatin	No significant reduction in steatosis
Vakili et al. [25]	2016	Normothermic	GDNF or PPARα and δ ligands, hypericin, scoparone, forskolin and visfatin	GDNF: equally effective as defatting agents at lowering TGs, and caused less liver damage (rise in LDH activity)
Raigani et al. [27]	2020	Normothermic	PPARα and δ ligands, hypericin, scoparone, forskolin, visfatin, L-carnitine and amino acids	Decreased perfusate lactate, better bile quality, and decreased inflammatory markers; increased β-oxidation markers; no significant reduction in steatosis

**Table 3 ijms-23-11805-t003:** Summary of Machine Perfusion Techniques–Clinical Trials.

First Author	Year	Temperature of Perfusion	Additional Agents	Effects of Perfusion
Guarrera et al. [28]	2010	Hypothermic	None	May improve graft function
Monbaliu et al. [29]	2012	Hypothermic	None	Discarded livers had higher levels of injury markers in the perfusate
Abudhaise et al. [30]	2018			
van Rijn et al. [31]	2021	HOPE	None	Less occurrence of biliary strictures, post reperfusion syndrome, and allograft dysfunction
Czigany et al. [32]	2021	HOPE	None	Lower levels of liver injury enzymes, graft dysfunction, 90-day complications, and hospital stay
Ravaioli et al. [33]	2020			
Watson et al. [34]	2018	Normothermic	None	Discarded livers were suitable for transplantation after perfusion
Mergental et al. [35]	2020			
Quintini et al. [36]	2022			
Liu et al. [37]	2018	Normothermic	None	Increased perfusate TG levels during treatment; no significant decrease in steatosis histologically
Boteon et al. [38]	2019	Normothermic	PPARα and δ ligands, hypericin, scoparone, forskolin and visfatin	Reduction in TGs and macrosteatosis, increased β-oxidation, higher ATP levels, and enhanced viability

**Table 4 ijms-23-11805-t004:** Summary of Genomic Approaches for Liver Defatting.

Gene	First Author	Year	Type of Modification	Effect of Genomic Modification
*FATP2*	Falcon et al. [41]	2010	Knockdown	Decreased lipid uptake and lower liver TGs
*FATP5*	Doege et al. [42]	2006	Knockout	Decreased lipid uptake, lower TGs, and reverses steatosis
	Doege et al. [43]	2008		
*CD36*	Wilson et al. [45]	2016	Knockout	Decreased lipid uptake, lower TGs, improved insulin sensitivity and reduced inflammatory markers
*Caveolin-1*	Li et al. [47]	2017	Knockdown	Increased steatosis, plasma cholesterol, and liver injury enzymes
	Li et al. [47]	2017	Overexpression	Decreased lipid accumulation
*FABP1*	Martin et al. [51]	2009	Knockout	Lower TGs and decreased lipid disposal pathways
	Mukai et al. [52]	2017	Knockout	Decreased expression of inflammatory markers
*SREBP1c*	Shimano et al. [55]	1997	Overexpression	Higher TG levels
	Liang et al. [56]	2002	Knockout	Decreased ACC and FASN mRNA levels (needed for DNL)
*ChREBP*	Lizuka et al. [57]	2004	Knockout	Lower TGs, but higher insulin resistance, delayed glucose clearance and simple sugar intolerance
	Zhang et al. [58]	2017	Knockout	Protection against steatosis but enhanced hepatic damage
	Benhamed et al. [60]	2012	Overexpression	Produced steatosis but maintained insulin sensitivity and glucose tolerance
*ACC1/ACC2*	Mao et al. [61]	2006	Knockout	Less production of malonyl-CoA, less TG accumulation; increased synthesis of lipogenic enzymes
	Savage et al. [62]	2006	Knockdown	Reversed steatosis, reduced malonyl-CoA, improved insulin sensitivity, and increased β-oxidation
	Matsumoto et al. [63]	2020	Small molecule inhibitors (ACC1/2)	Lower TGs, reduced fibrosis and lowered liver injury markers; higher plasma TGs
	Tamura et al. [64]	2021	Small molecule inhibitors (ACC1)	Reduction in steatosis and fibrosis; no change in plasma TGs
*FASN*	Li et al. [65]	2016	Knockout	Hypoglycemia, liver steatosis, and decreased β-oxidation; decline in cell proliferation and rise in apoptosis
*SCD1*	Li et al. [66]	2009	Knockout	Increased fibrosis and cellular apoptosis
*ZHX2*	Zhao et al. [67]	2022	Overexpression	Stopped progression of steatosis and reduced liver inflammation
			Knockout	Increased lipid accumulation, increased fibrosis and enhanced liver inflammation

## Data Availability

Not applicable.

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
