# Peer review of "A Review of Defatting Strategies for Non-Alcoholic Fatty Liver Disease"

_ijms, 2022, doi:10.3390/ijms231911805_

Round 1
Reviewer 1 Report
The paper is an interesting review about the techniques of defatting livers, with a more in depth attention to the field of transplantation.
I have only few suggestion for the authors.
- I aknowledge that this is not a systematic review, however it would have been helpful to know the criteria by which the references had been found and cited
- I think that tables describing the studies for each section (in vitro, machine perfusion, genomic approaches) would be valuable to help the reader recap the information from each paragraph
- A study has been published this year on Hepatology about the role of hyperexression of hepatocyte zink finger and homeboxes 2 (ZHX2) in alleviating NASH. Probably this could be another therapeutical target for NASH treatment which should be mentioned.
Author Response
I acknowledge that this is not a systematic review, however it would have been helpful to know the criteria by which the references had been found and cited.
As reviewer-1 predicts, this is not a systemic review. Our criteria to choose the reference papers is based on the IJMS (MDPI) regulations. We try to include 50% of our references from recent studies which is published in last 5 years and try to find the most impactful papers in the field of liver defatting. We screened the field of defatting liver studies in diverse research background including molecular biology, chemical treatments, current ongoing medical applications and potential techniques.
- I think that tables describing the studies for each section (in vitro, machine perfusion, genomic approaches) would be valuable to help the reader recap the information from each paragraph.
Thanks for the review comment, this idea is very useful. We included separate tables to summarize each section.
- A study has been published this year on Hepatology about the role of hyperexression of hepatocyte zink finger and homeboxes 2 (ZHX2) in alleviating NASH. Probably this could be another therapeutical target for NASH treatment which should be mentioned.
Thanks for this valuable contribution. We already include ZHX2 as a separate paragraph in the main text and included also in the main figure2, panel3.
Reviewer 2 Report
It is interesting review regarding therapeutic approach for liver defatting. The paragraph on the effects of bariatric surgery as well as glucagon-like peptide 1 agonists is in the opinion of the reviewer missing in this study. However, apart from that, the study is thoroughly written. The last remark: the title of the study is about liver defatting in general, and the conclusions are mostly about NAFLD.
Author Response
It is interesting review regarding therapeutic approach for liver defatting.
Thanks for this comment.
The paragraph on the effects of bariatric surgery as well as glucagon-like peptide 1 agonists is in the opinion of the reviewer missing in this study.
We appreciate reviewer2 for reminding us the very important missing point of this review paper. Bariatric surgery needs to be discussed as an independent treatment approach. Therefore, we wrote another section at the end of the manuscript as an additional intervention method to summarize the role of bariatric surgery and GLP-1 for defatting. We also include bariatric surgery as a separate panel in Figure 2.
However, apart from that, the study is thoroughly written. The last remark: the title of the study is about liver defatting in general, and the conclusions are mostly about NAFLD.
As review2’s suggestion, new title becomes “A Review of Defatting Strategies for Non-Alcoholic Fatty Liver Disease”.